# Transcontinental dispersal of *Anopheles gambiae* occurred from West African origin via serial founder events

Hanno Schmidt [1], Yoosook Lee[1], Travis C. Collier [1], Mark J. Hanemaaijer[1], Oscar D. Kirstein[1], Ahmed Ouledi[2], Mbanga Muleba[3], Douglas E. Norris[4], Montgomery Slatkin[5], Anthony J. Cornel[1,6] & Gregory C. Lanzaro[1]*

The mosquito *Anopheles gambiae s.s.* is distributed across most of sub-Saharan Africa and is of major scientific and public health interest for being an African malaria vector. Here we present population genomic analyses of 111 specimens sampled from west to east Africa, including the first whole genome sequences from oceanic islands, the Comoros. Genetic distances between populations of *A. gambiae* are discordant with geographic distances but are consistent with a stepwise migration scenario in which the species increases its range from west to east Africa through consecutive founder events over the last ~200,000 years. Geological barriers like the Congo River basin and the East African rift seem to play an important role in shaping this process. Moreover, we find a high degree of genetic isolation of populations on the Comoros, confirming the potential of these islands as candidate sites for potential field trials of genetically engineered mosquitoes for malaria control.

[1] Vector Genetics Laboratory, Department of Pathology, Microbiology and Immunology, School of Veterinary Medicine, University of California - Davis, Davis, CA 95616, USA. [2] Université des Comores, Grande Comore, Union of the Comoros. [3] Tropical Disease Research Centre, Ndola, Zambia. [4] The W. Harry Feinstone Department of Molecular Microbiology and Immunology, The Johns Hopkins Malaria Research Institute, Johns Hopkins Bloomberg School of Public Health, Baltimore, MD 21205, USA. [5] Department of Integrative Biology, University of California - Berkeley, Berkeley, CA 94720, USA. [6] Mosquito Control Research Laboratory, Department of Entomology and Nematology, University of California - Kearney Research and Extension Center, Parlier, CA 93648, USA. *email: gclanzaro@ucdavis.edu

Evolutionary processes within species depend on the spatial and genetic connectivity of their populations. This is especially true when considering large spatial scales and long time periods, i.e., the full geographical and historical space of a species[1,2]. The various processes underlying these population dynamics can be described at the organismal level as migration of individuals between populations or on the genetic level as ancient and recent gene flow between them. Gene flow essentially reflects the direct exchange of heritable information due to migration between related populations in an evolutionary sense[3]. Barriers to migration play a pivotal role in shaping patterns of inter-population gene flow. These comprise different elements of the landscape such as physical (e.g., roads[4,5]) and ecological barriers (i.e., unsuitable habitats[6]), with the latter often being species-specific and erratic[7]. The most obvious and universal barrier to migration is realized in islands, hence the term "island biogeography"[8]. Isolation of island populations depends on the dispersal ability of the species, the remoteness of the island, and various external factors like sea currents[9], wind directions[10], and anthropogenic activity[11]. Knowledge about the connectivity between island and mainland populations can help to facilitate an understanding of how newly introduced genetic variants can spread throughout the distribution area of a species.

Anopheles gambiae s.s. GILES (referred to as A. gambiae henceforth) is a common mosquito occurring throughout sub-Saharan tropical Africa[12]. It is of high importance in public health for its role as the principal vector of human malaria, a disease causing >200 million cases and half a million deaths per year according to World Health Organization (WHO) estimates[13]. A recent modeling effort suggests that malaria is not expected to be eliminated with currently available tools in highly endemic areas[14,15] due to lack of a vaccine[16], increasing drug resistance[17], spreading insecticide resistance alleles[18], and decreasing effectiveness of physical prevention provided by bed nets[19]. Consequently, renewed efforts to explore the use of genetically engineered mosquitoes (GEMs) for malaria control have been initiated[20–23], including the development of models to explore the dispersal capacity of gene drive systems through natural populations[24–26]. Knowledge of the varying degrees of connectivity between different populations throughout the species' range as well as nucleotide diversity that could provide a potential mechanism for gene drive resistance[27,28] therefore will be important for assessing the outcome of field release trials of GEM or conventional control measures.

The importance of A. gambiae has led to extensive studies linking patterns of genetic diversity to different ecological conditions[29,30] as well as diverse population genomics studies[31,32]. Recently, a broad-scale population genomic analysis across most of continental Africa revealed pronounced structure and varying levels of gene flow among geographic populations[33]. However, oceanic island populations were not included in this study. A number of individual studies of island populations of A. gambiae have been undertaken, most recently analysis of several small islands in Lake Victoria, eastern Africa, indicating genetic isolation of these island populations[34]. However, these islands are located <50 km from the mainland and there is a known negative correlation between species diversity and distance to mainland[35] including a study specifically on mosquitoes[36], underlining the importance of remoteness for isolation.

The Comoros form an archipelago consisting of four islands situated in the Mozambique Channel, approximately 300 km off the African coast and 300 km north-west of Madagascar. An initial study comparing Comorian and mainland African A. gambiae populations suggest a high degree of genetic isolation[37]. Here we present individual whole-genome resequencing data for populations from the three Comoro islands Grande Comore (Comorian: Njazidja), Anjouan (Nzwani) and Mohéli (Mwali) that comprise the Union of the Comoros, as well as populations from four continental African countries. We chose Mali as a representative of the core distributional area of A. gambiae in western Africa plus sites in Cameroon, Tanzania, and Zambia, which span sub-Saharan Africa (Fig. 1). This study is the first whole-genome population genetic analyses of A. gambiae across Africa that includes remote island populations with the goal of generating a more inclusive view of the species' evolutionary history.

## Results

**Sequencing, pre-processing of sequence data, mapping, and variant calling.** Sequencing of the 111 samples resulted in 3.7 billion reads in total with a mean genome coverage of 11.7× per sample (Supplementary Table 1). On average, 95.6% of the reads were mapped onto the reference genome AgamP4[38] by BWA-MEM[39]. Applying joint variant calling with Freebayes[40], we identified 24,069,835 high-quality biallelic single-nucleotide polymorphisms (SNPs) in the dataset.

**Intra- and inter-population genetic variability.** Genetic variability was estimated for each population by dividing the number of biallelic SNP sites in the heterozygous state by the total number of loci. Genetic variability was highest in west Africa (Mali, Cameroon), intermediate in east Africa (Zambia, Tanzania), and lowest for the three Comoro islands (Fig. 2). Plotting genetic structure in a spatial framework with SpaceMix[41] resulted in one large cluster of samples from Mali and an adjacent, partially overlapping cluster of the Cameroon samples (Fig. 3). The remaining samples distinctly clustered by geographical location (Tanzania, Zambia, Grande Comore, Mohéli, Anjouan). Geographic distances were not correlated to geogenetic distances. For example, the geogenetic distances between the two smaller islands of the Comoros, Mohéli and Anjouan, and Grande Comore as well as between each of the islands of the Comoros and Tanzania was greater than the distance between Cameroon and Mali (Fig. 3). Consistent with these results, the dispersal capacity $N_x\sigma^2$ estimated using the Raddle[42] software on rare allele distribution were almost four times higher in Mali (2342) than within the populations of the Comoros (623).

Hudson's $F_{ST}$[43,44] as a measure of relative population differentiation was estimated for each population pair (Supplementary Table 2) and combined into a phylogenetic tree (Fig. 4a) to display broader patterns. Bootstrap values were >57 for every node and at least 98 for all major ones. While all Mali populations form a single cluster with short branches, the populations from the three Comoro islands form three well-defined clusters with the branches from the two smaller islands being closer to each other than to the branch of Grande Comore. The remaining three mainland sites were located in between, with Cameroon being closest to Mali, Tanzania closest to the Comoro islands, and Zambia between them with less genetic distance to Tanzania than to Cameroon. Divergence among populations within Mali was low and independent of geographic distance (Fig. 4b). Divergence between populations located within each of the Comoro islands was likewise low and not correlated with geographic distance, which is not surprising given the small size of the islands. However, $F_{ST}$ values between islands revealed much higher levels of divergence. Also, $F_{ST}$ values between populations from the islands compared with the mainland were consistently higher than between different mainland sites (Fig. 4c). Overall, the distribution of $F_{ST}$ values suggests a low impact of geographical distance on population differentiation and a moderate-to-high degree of divergence between populations from the Comoros and mainland populations.

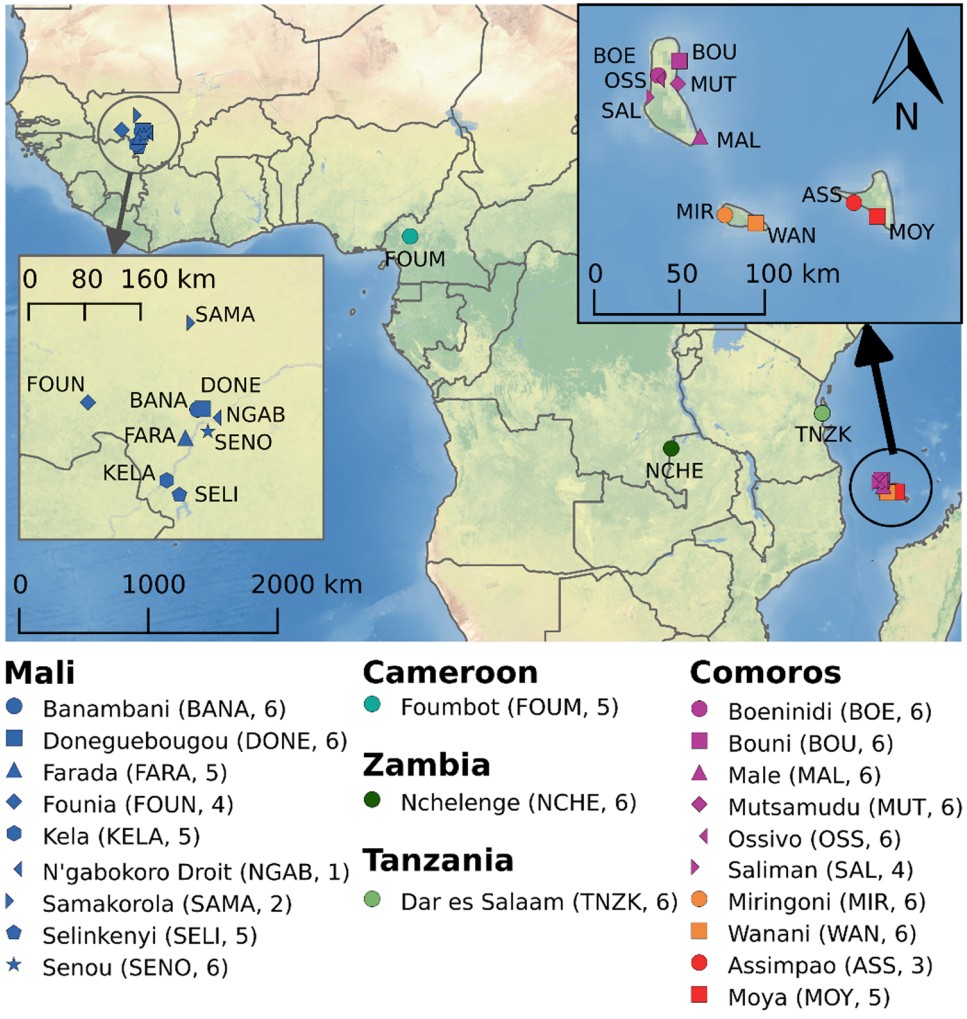

**Mali**
- ● Banambani (BANA, 6)
- ■ Doneguebougou (DONE, 6)
- ▲ Farada (FARA, 5)
- ◆ Founia (FOUN, 4)
- ⬡ Kela (KELA, 5)
- ◀ N'gabokoro Droit (NGAB, 1)
- ▶ Samakorola (SAMA, 2)
- ⬠ Selinkenyi (SELI, 5)
- ★ Senou (SENO, 6)

**Cameroon**
- ● Foumbot (FOUM, 5)

**Zambia**
- ● Nchelenge (NCHE, 6)

**Tanzania**
- ● Dar es Salaam (TNZK, 6)

**Comoros**
- ● Boeninidi (BOE, 6)
- ■ Bouni (BOU, 6)
- ▲ Male (MAL, 6)
- ◆ Mutsamudu (MUT, 6)
- ◀ Ossivo (OSS, 6)
- ▶ Saliman (SAL, 4)
- ● Miringoni (MIR, 6)
- ■ Wanani (WAN, 6)
- ● Assimpao (ASS, 3)
- ■ Moya (MOY, 5)

**Fig. 1 Collection sites.** Samples were collected in Mali (dark blue), Cameroon (turquoise), Zambia (dark green), Tanzania (light green), and the Comoros (purple for Grande Comore, orange for Mohéli, red for Anjouan). A map of the three islands of the Comoros enlarged for details is shown in the upper right corner. The CleanTOPO2 basemap was used as background.

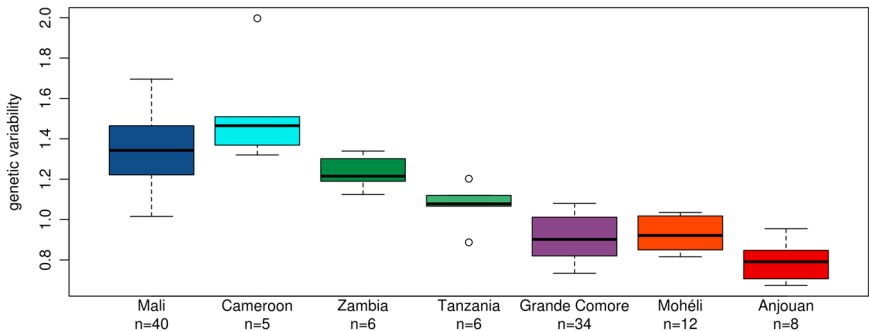

**Fig. 2 Genetic variability.** Genetic variability was calculated for every sample individually as the number of biallelic SNP sites in heterozygous state divided by the total number of loci and given as a percentage. The number of samples per group is given under each location name.

**Admixture of populations**. Population ancestry patterns were explored with ADMIXTURE[45] for seven scenarios with two to eight assumed ancestral populations (Fig. 5, Supplementary Fig. 1). Results with two assumed ancestral populations ($K = 2$) show a separation between west (Mali, Cameroon) and east (Zambia, Tanzania and the Comoros) African populations. With $K = 5$, the samples from the Comoro populations are assigned to three genetic clusters, two present in Grande Comore and the third corresponding to all individuals from the two smaller

islands Mohéli/Anjouan. One of the genetic clusters in Grande Comore is also present in East African populations (Zambia/ Tanzania), suggesting closer genetic relationship between the main island and mainland than between the smaller islands and mainland (as depicted in the $F_{ST}$ tree (Fig. 4a)). West African samples are assigned to two clusters with one being more prevalent in samples from Mali and the other in Cameroon. Zambia/ Tanzania shows an intermediate state between the genetic cluster of Cameroon and Grande Comore. With the assumed ancestral

populations increased to seven ($K = 7$), the three Comoro islands are distinct and separate from Tanzania and Zambia, which themselves form a single group. Cross-validation error analysis revealed $K = 5$ as the best fit (Supplementary Fig. 2); however, $K = 7$ appears to be more consistent with ancestry relationships described in earlier reports[37] and with the $F_{ST}$ tree and SpaceMix analyses presented in this paper, all of which suggest high divergence between the island and east African populations and among the three Comoro islands themselves.

**Historical population sizes and cross-coalescence.** Historical effective population sizes estimated by multiple sequentially

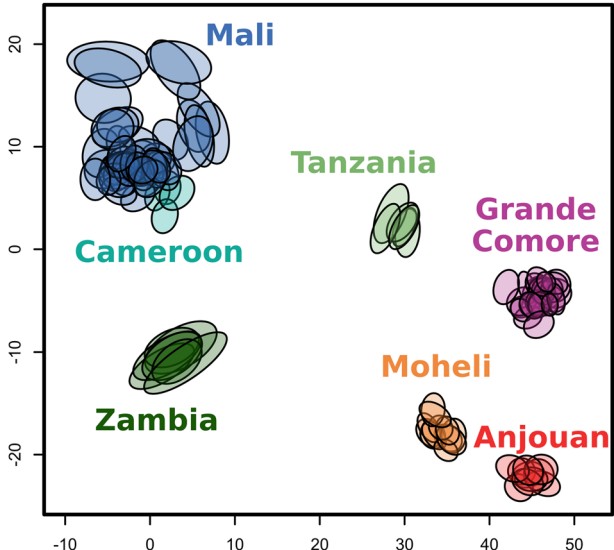

**Fig. 3 SpaceMix results.** SpaceMix-inferred geogenetic locations of samples based on prior of true sampling sites and population consolidated SNP data[41]. *X* and *Y* axes are coordinates in a geogenetic space, where space is warped to place the genetically similar individuals together. An ellipse indicates an individual mosquito sample. The area of each ellipse represents the 95% CI for location where an individual could have originated in geogenetic space.

Markovian coalescence using MSMC2[46] follow a similar trajectory in all populations in the deep past (Fig. 6; deep to recent past from right to left). However, ~200,000 years ago we see a clear split into two groups with west African populations (Mali, Cameroon) on one trajectory and east African populations (Zambia, Tanzania and the Comoros) on another. West African population sizes continue to remain high and relatively constant following that first split. Eastern populations undergo a series of splits, the first separating Zambia from the others, a second separating the Comoros from Tanzania, and finally a third separating the islands of Anjouan and Mohéli from the island of Grande Comore. Eastern populations, however, experience a drop in population size, which suggests founder effect (s)[47,48]. While the Zambian population subsequently increases and levels off, the others (Tanzania, Comoros) continue to decline. Somewhat later, ~40,000 years ago, the Tanzanian population increases while the population sizes in the Comoros continue to decline. The islands of Anjouan and Mohéli show similar patterns, reaching their minimal turning point at about the same time with the island of Grande Comore, increasing slightly later. Approximately 25,000 years ago, all populations have reached or passed their minimal turning point and are increasing in size again.

Comparing the distribution of SNPs between samples can provide an estimate of the shared history of populations in the form of cross-coalescence[49]. A higher relative cross-coalescence (RCC), calculated using the MSMC2 algorithm, indicates less time to the last common ancestor shared by the two populations in a specific comparison. All populations shared common ancestors in the deep past, reflecting high connectivity, probably as one super-population. Around the time when we see the potential initial founder effect (Fig. 6a), we also see decrease of RCC between all comparisons except the two Mali populations (Fig. 6b). Most recent reliable estimates of RCC that include all pairwise comparisons could be made at ~30,000 years ago. These are illustrated as a cross-sectional slice through the curves, as shown in Fig. 6b. Several hypothesis bearing on the relationships among populations can be formulated from these results, including RCC between different populations in Mali is almost one, equivalent to nearly no population structure; RCC between west African populations (Mali, Cameroon) declined slightly and stabilized at a high level (~0.7), indicating substantial gene flow; RCC between the two smaller Comoro islands (Mohéli and Anjouan) is comparable to that between Mali and Cameroon,

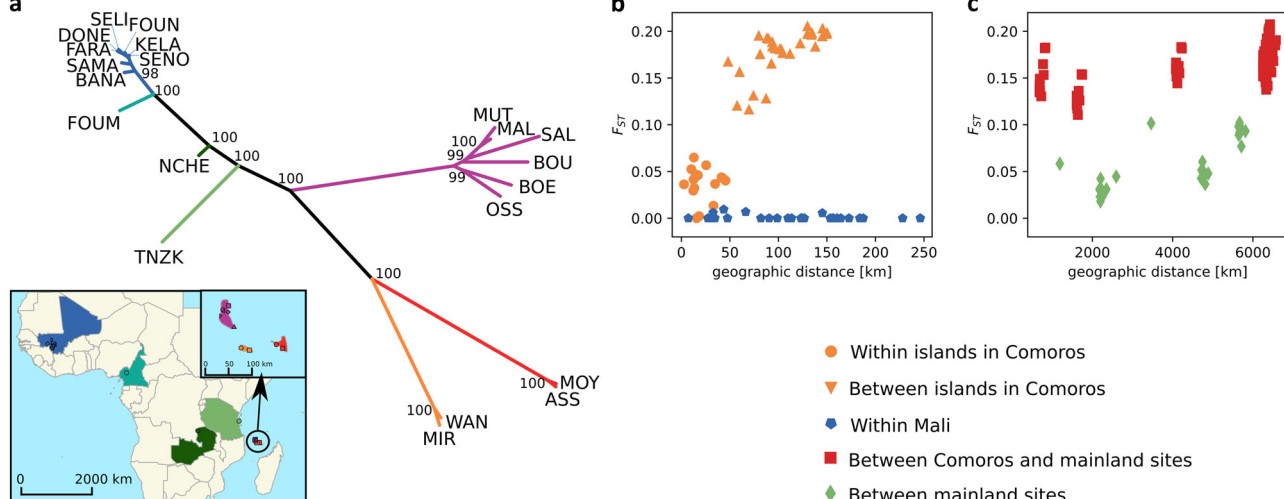

**Fig. 4 $F_{ST}$ analyses. a** Unrooted tree based on pair-wise $F_{ST}$ values using Neighbor-joining algorithm. $F_{ST}$ values were calculated for all pairwise comparisons. Long branches separate population pairs with high $F_{ST}$ estimates and topology reflects similar overall patterns of $F_{ST}$ values toward other populations. Of note, all branches had bootstrap values >57. Owing to tight spacing between nodes, only the bootstrap values ≥98 are displayed in the figure. See Fig. 1 for color scheme and site key. **b**, **c** Correlation between geographic and genetic distance ($F_{ST}$) at various spatial levels.

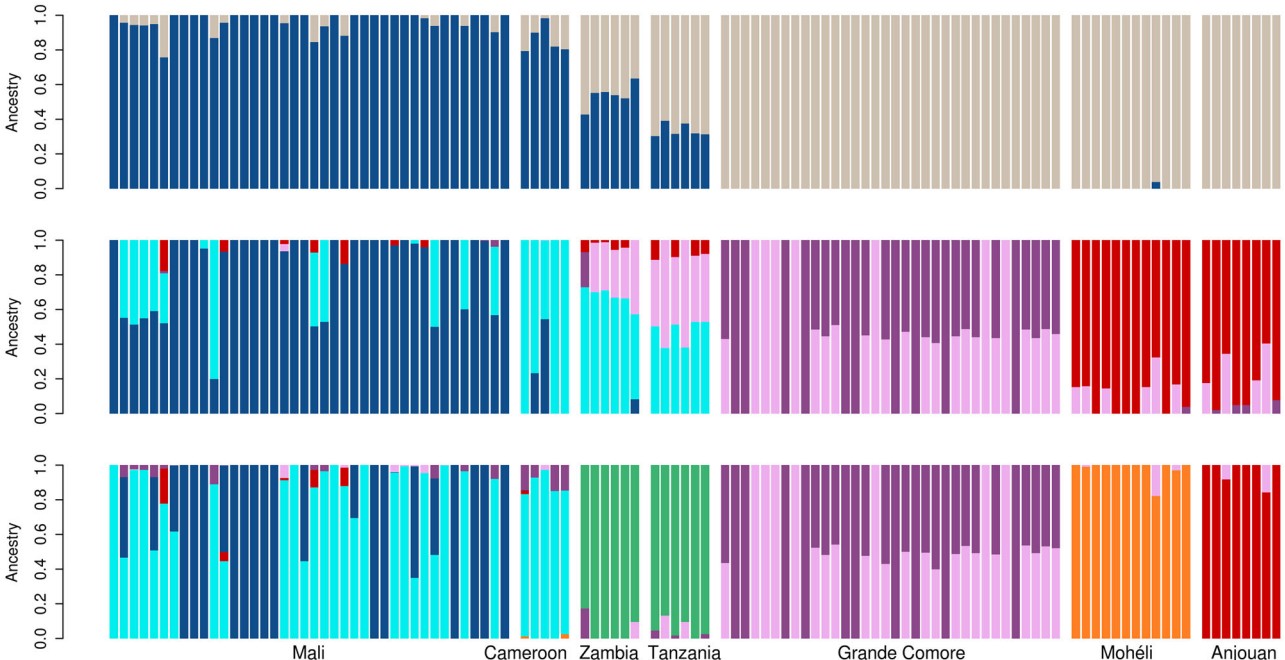

**Fig. 5 Admixture analysis.** SNP data are used to estimate individual ancestries, and thereby population structure. Shown are results for two, five, and seven assumed ancestral populations. Samples were grouped by location (see labels at bottom). The cross-validation error analysis[45] revealed $K = 5$ as the best fit (see Supplementary Fig. 2).

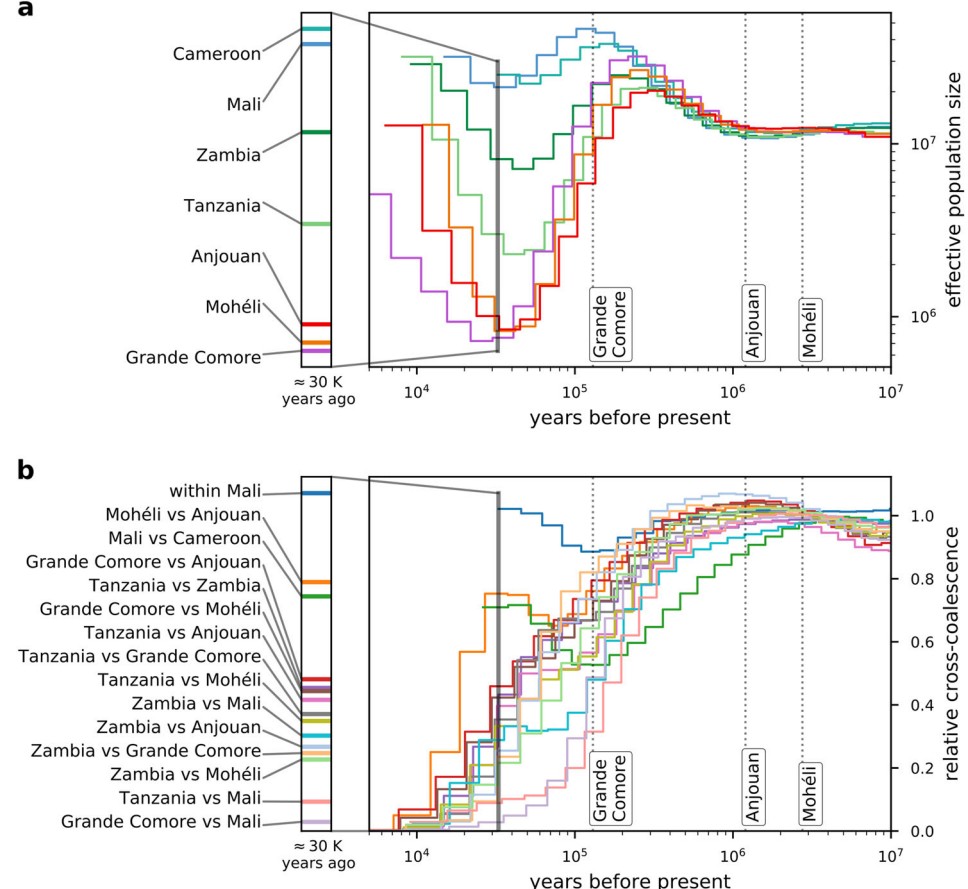

**Fig. 6 Historical effective population sizes and coalescence estimates. a** Historical effective population sizes. **b** Relative cross-coalescence (RCC) between populations. The dotted lines depict the geological appearance of the three Comoro islands[55]. The vertical bar marks the time point most closely to present where estimates for all populations are available and is shown enlarged at the left. Legend entries are given in the same order as they appear in the bar. Note the reading direction "deep to recent past" from right to left and the logarithmic scales in plots (all but y axis in **b**).

thereby suggesting distinct populations but with some degree of historical and possibly contemporary gene flow; RCC between west and east African populations declines steadily over time, with Tanzania and the Comoros becoming increasingly isolated from Mali/Cameroon; RCC is lowest between Tanzania and Mali and Grande Comore and Mali (reaching ~0.2 ~100,000 years ago), indicating very strong isolation between west African and these eastern-most African populations; RCC between Comorian and continental populations (Tanzania, Zambia) is quite low, comparable to between Zambia and Mali, indicating strong and enduring isolation (Fig. 6b, Supplementary Fig. 3).

## Discussion

The population genomic analyses described here reveal several patterns that have major implications for our understanding of the population biology and historical phylogeography of *A. gambiae*. Most strikingly, geographic distances are not correlated to genetic distances between populations. For example, geogenetic locations of west African populations (Mali, Cameroon) are inconsistent with their true geographic location (Fig. 3). The two groups share the same geogenetic space despite being located roughly 2500 km apart (Fig. 1). These results highlight the potential for historical gene flow to maintain population homogeneity over time, even between populations located at distant sites and in a species with relatively low vagility[50]. The pattern is quite different for eastern populations (Tanzania, Zambia) that are widely separated in both geogenetic and geographic space (separated by 1200 km). This pattern is even more prominent within the Comoros, which are geographically proximal (~40 km). Grande Comore is clearly separated from Mohéli and Anjouan in geogenetic space (Fig. 3), $F_{ST}$-based tree topology (Fig. 4a), and Admixture analyses (Fig. 5). These findings underscore the importance of geographic features that can function as barriers to population connectivity, i.e., gene flow in *A. gambiae*.

The geographical origin of *A. gambiae* was previously hypothesized to be located in eastern[51] or central Africa[52,53]. From our analyses of the historical population sizes and cross-coalescence, it seems more likely that *A. gambiae* originated in western Africa. West African populations do not appear to have experienced strong historical fluctuations in population size, whereas east African populations each experienced a dramatic decrease in effective population size, followed by a steady increase (Fig. 6a). This could be explained by a series of population subdivisions involving the establishment of new geographic

populations from a small portion of the ancestral population (founder effect), resulting in the species' range expanding from west to east. We hypothesize that these founder populations were initiated with a small number of individuals crossing a geological barrier that limits gene flow. The first of these events occurred ~200,000 years ago resulting in a split between west African (Mali, Cameroon) and east African (Tanzania, Zambia, Comoros) populations, resulting in newly founded east African populations showing reduced population size (Fig. 6a) and a subsequent decline in RCC (Fig. 6b). The geological barrier involved may be the Congo River basin, which has been shown to form an effective barrier to gene flow in birds[54]. The next split occurred ~100,000 years ago between Zambia and the other east African (Tanzania, Comoros) populations, with steeper subsequent decline in population size for the latter group. This event may be attributed to crossing the East African rift, which has previously been shown to be associated with strongly increased $F_{ST}$ values in *A. gambiae*[33], hence suggesting a hurdle for gene flow. Presumably, the Comoros were colonized by founders migrating across the Mozambique Channel from easternmost continental African populations. Our data suggest that roughly ~70,000 years ago the Comoros *A. gambiae* population split from Tanzania, followed by another split ~40,000 years ago between Grande Comore and Mohéli and Anjouan. That Grande Comore was the most recently colonized is supported by the fact that it is the youngest of the islands geologically[55] and is the latest to have experienced a dramatic increase in population size (Fig. 6a). Closer genetic relationship of Grande Comore to mainland sites is supported by our $F_{ST}$ (Fig. 4) and Admixture (Fig. 5) analyses, which alternatively makes independent colonization from the mainland seem possible. A steady decline in genetic variability from west to east African populations (Fig. 2) and the clear separation of east African (Tanzania, Zambia, Comoros) from west African populations (Mali, Cameroon) in geogenetic space (Fig. 3) support the idea of a west African origin of the species, followed by episodic migrations eastwards. Although we cannot rule out slightly different scenarios, the proposed model (Fig. 7) seems to be the more parsimonious interpretation of our data.

Our data indicate that *A. gambiae* likely became established on the Comoros hundreds of thousands of years after their geological formation[55] but most likely prior to permanent human settlement ~1300 years ago[56]. These founders must have fed on other hosts (e.g., bats or birds) prior to human habitation. This would be

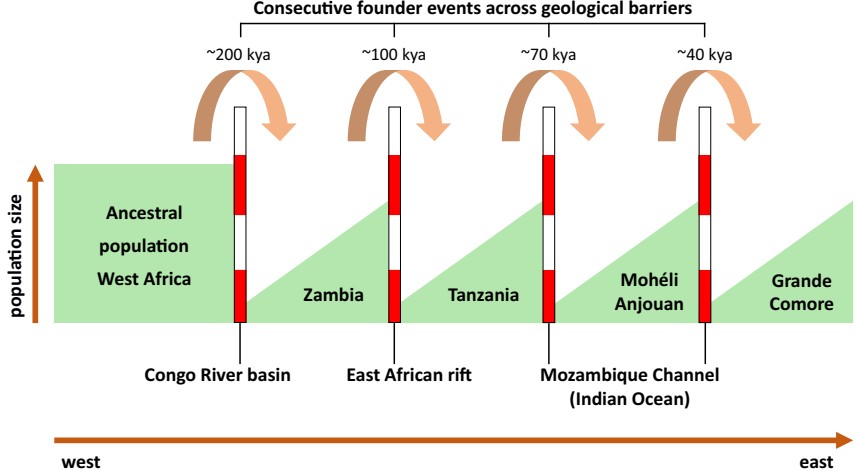

**Fig. 7 Model of the species' dispersal across Africa.** Based on our data, we hypothesize an origin of *A. gambiae* in west Africa with dispersal eastwards. In this process, the geological barriers Congo River basin, East African rift, and Mozambique Channel interrupted the species' unhindered expansion. This led to a series of founder events with subsequent increase in population size at the new location after a small number of individuals crossed the barrier. Kya = thousand years ago.

consistent with the assumption that *A. gambiae* adapted only recently (~10,000 years ago) to humans as hosts[57,58] and was not restricted to non-human primates before[59].

Our study has some minor bias regarding sampling of specimens at different sites. For instance, our 2011 Comoros collection is composed of larvae only since we were not able to locate indoor-resting adults. It has been shown in Goundry, Burkina Faso that larval collections contained hybrid populations that are distinct from typical *A. gambiae* or *Anopheles coluzzii* adult collections from the same village[60]. We minimize this potential bias by genotyping Divergence Island SNPs (DISs)[61] prior to whole-genome sequencing. This assay provides a more in-depth genetic background characterization than traditional PCR-based methods and can distinguish hybrid from parental form individuals[62,63]. All Comoros samples had typical *A. gambiae* genotypes[64]. The temporal dynamics of *A. gambiae* populations, their resting/biting behavior, and ideal method of adult surveillance are some of areas that need further study.

Our data may contribute to efforts currently underway to evaluate the deployment of genetically engineered *A. gambiae* for malaria control in Africa[65,66]. Identification of confined field trial sites aimed at evaluating the performance of GEMs includes measures of gene flow between target and off-target sites. Our results indicate that gene flow is extensive among mainland sites but highly restricted between mainland and sites on oceanic islands. The Comoro Islands have relatively small *A. gambiae* populations (Fig. 6a) that are relatively isolated both genetically and physically from mainland Africa (Figs. 3, 4c, 5, and 6b). Moreover, the island of Grande Comore is isolated from the other two islands that make up the Union of the Comoros, as illustrated by its distance in geogenetic space being comparable to that between Tanzania and Grande Comore (Fig. 3), $F_{ST}$ branch lengths (Fig. 4a), and distinct genetic clusters in Admixture analyses (Fig. 5). This feature of the genetics of these populations presents the possibility of performing staged and/or parallel trials with GEMs. This could be especially useful since there would be two distinct systems: the two smaller islands including populations with a strong but possibly permeable barrier separating them, and Grande Comore with an isolated but more structured population (Fig. 5). Further studies will be needed, such as estimating recent population size using identity-by-descent sharing[67] to identify recent migrations and to describe the genetic structure of populations within each island. We conclude that mosquito populations on isolated oceanic islands, such as the Comoros, could make ideal sites for conducting ecologically contained field trials of GEMs, following the guidelines set out by the WHO[68].

## Methods

**Mosquito samples.** We used $N = 111$ *A. gambiae* specimens ($N = 40$ from Mali, $N = 5$ from Cameroon, $N = 6$ from Tanzania, $N = 6$ from Zambia, and $N = 54$ from the Comoros) from the Vector Genetics Laboratory archive mosquito DNA collection for the study (Fig. 1, Supplementary Table 3). Samples from Mali and Cameroon were collected as female adults inside houses using mouth aspirators in August 2006. Samples from the Comoros were collected as larvae inside cisterns using scoops and transfer pipets in February 2011[62]. Tanzania samples were collected in 2012 and Zambia samples in April and May 2015 as adults by pyrethroid spray catch collection. Species identity was confirmed using the DIS assay described in ref. [69]. Sex of larvae was determined by a Y chromosome PCR method[70] and confirmed by examining the ratio of coverage of Y_unplaced contig relative to nuclear genome coverage. For females, the median Y_unplaced to nuclear genome coverage ratio was 1.00. Any specimens exceeding the Y/nuclear genome ratio of 10 were not included in this study. This ensures that all samples we analyzed are females. Only samples that showed *A. gambiae*-specific genotypes for all 15 loci screened were used for whole-genome sequencing.

**Whole-genome sequencing.** DNA extraction was performed using the protocol described in refs. [71,72]. DNA concentrations of the samples was measured using *dsDNA HS Assay Kits* on a Qubit instrument (Life Technologies). Individual genomic DNA libraries were constructed using 10 ng DNA, the KAPA HyperPlus

Kit (KAPA Biosystems), custom IDT 48 dual index barcodes (Integrated DNA Technologies), and Ampure SPRI beads (Beckman). We applied the library construction protocol described in ref. [72]. Library concentrations were measured using Qubit as described above. Individually barcoded libraries were combined by equal quantity for pooled sequencing based on the Qubit results. Sequencing was performed using an Illumina HiSeq 4000 instrument at the UC Davis DNA Technologies Core.

**Pre-processing of sequence data, mapping, variant calling.** Demultiplexed raw reads were filtered and trimmed (ILLUMINACLIP:2:30:10, LEADING:3, TRAILING:3, SLIDINGWINDOW:4:15, MINLEN:36) using Trimmomatic v0.36[73]. PCR duplicates were removed using Picard Tools v2.2.4 (http://broadinstitute.github.io/picard/) and reads were realigned around indels with GATK v3.5[74]. Afterwards the processed reads were mapped to the reference genome *Agam*P4[38] using BWA-MEM v0.7.15[39] with default settings. Freebayes v1.0.1[40] was used for variant calling, applying default parameters but "theta = 0.01" and "max-complex-gap = 3." Variants without support from both overlapping forward and reverse reads were removed. Only biallelic SNPs with minimum depth of 8 were called for genotypes and used for analysis.

**Intra- and inter-population genetic variability.** The genetic variability was calculated for each mosquito as the number of biallelic SNP sites in heterozygous state divided by the total number of loci in the genome[75]. Including all SNPs would have arbitrarily weighted the similarity/difference to the reference genome instead of the variability that is indicative of population size and history[76]. The data were plotted grouped by origin using the boxplot function in R v3.2.5[77] applying default settings (Tukey boxplot). The spatial patterns of population genetic structure were analyzed using the R package SpaceMix v0.13[41] to create a geogenetic map of the polymorphism data. In these, allele frequency covariance is displayed in a way that distances between samples reflect geogenetic, rather than geographic, distance. Neighborhood sizes $N_s\sigma^2$, which give an estimate of dispersal distances, were estimated using Raddle[42] with settings recommended by the authors (https://github.com/NovembreLab/raddle). Raddle searches for rare alleles in the data and calculates dispersal capacities by their distribution within and across populations.

As another measure of population differentiation due to genetic structure, pairwise fixation indices $F_{ST}$ were calculated using Hudson's estimator, which is not sensitive to the ratio of sample sizes and does not systematically overestimate $F_{ST}$[44], implemented in scikit-allel v1.2.0[78] utility package. As a proxy for relatedness of the populations in terms of relative genetic variance, a phylogenetic tree was calculated from the $F_{ST}$ data using a Neighbor-Joining[79] algorithm implemented in the program Neighbor (distance matrix model F84) from the package PHYLIP v3.696[80]. To estimate robustness of the topology, we re-calculated the tree 100 times with different subsets of randomly chosen 50K SNPs each for the distance matrix and derived the consensus rate in each branching point.

**Admixture profiles of populations.** Populations were screened for their admixture profiles using ADMIXTURE v1.3.0[45]. Admixture analyses were performed for a broad range of assumed ancestral populations ($K = 1$–10) and analyzed and plotted in R. The best-fitting $K$ was determined by ADMIXTURE's cross-validation procedure and the resulting error values.

**Historical population sizes and cross-coalescence.** The VCF files were filtered with VCFtools v0.1.12b[81] to remove samples with >5% missing data (i.e., SNPs not called) and sites with >5% missing data (i.e., SNPs not called). Afterwards the variants of all specimens were phased into haplotypes using SHAPEIT2 v2.9[82]. Phasing was done for 2L, 2R, 3L, and 3R separately (the X chromosome was not used here since recombination is different in sex-determining regions[83]). The SHAPEIT output was converted to VCF files and samples were separated again. Of those, only four samples per population (when possible) were used for the estimation of population sizes ($N = 83$) and two per population for the inter-population cross-coalescence, respectively (haplotypes $N = 8$ each), due to high computational demands. Reliable estimations can be calculated from haplotype numbers as low as 2 or 4 due to the mosaic nature of recombining nuclear genomes[46].

Estimation of historic effective population sizes $N_e$ and historic cross-coalescence was performed using the multiple sequentially Markovian coalescent pipeline MSMC2 v2.0.2[46] following the guide (https://github.com/stschiff/msmc2). For the MSMC2 analyses, mappability masks were prepared for the reference genome following the procedure of Heng Li's SNPable program (http://lh3lh3.users.sourceforge.net/snpable.shtml). In addition, mask files for the specimen's VCF files were produced using BEDOPS v2.4.30[84]. The MSMC2 main runs were conducted separately for each population to estimate within-population coalescence and in addition between a selected set of populations for cross-population coalescence with 20 Baum–Welch iterations each. The results were then converted to real time in years (assuming ten generations per year), population sizes, and RCC following the procedure published by the authors of MSMC2 (https://github.com/stschiff/msmc/blob/master/guide.md). A mutation rate of $2.85 \times 10^{-9}$ was assumed for the conversion, which is the median of the published mutation rates from the insect species *Drosophila melanogaster* (2.8e−9[85]), *Heliconius melpomene* (2.9e−9[86]), *Chironomus riparius* (2.1e−9[87]), and *Bombus*

*terrestris* ($3.6e-9$[88]). All time slices with $\lambda < 5$ were excluded to account for possible phasing error affecting resolution[89].

**Statistics and reproducibility**. One hundred and eleven specimens from 22 populations from west to east Africa were used for this study. Statistical analyses were performed in R and Python with programs cited or custom scripts provided under "Code availability." All tests and corresponding *p* values are reported in the text. $F_{ST}$ tree topology was tested by bootstrapping (100 times with different subsets of randomly chosen 50K SNPs each). Coalescence estimates with MSMC2 were generated with 20 Baum–Welch iterations each.

**Reporting summary**. Further information on research design is available in the Nature Research Reporting Summary linked to this article.

## Data availability

Sequence data that support the findings of this study are deposited in NCBI GenBank with accession numbers SAMN13337315–SAMN13337425 under BioProject ID PRJNA590708.

## Code availability

Codes used for analysis are available on GitHub page: https://github.com/travc/gambiae-dispersal

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

## Acknowledgements

We thank Kija N'ghabi (Ifakara Health Institute) for providing samples from Tanzania. We thank Lutz Froenicke and his team at the UC Davis DNA Technologies Core for carrying out genome sequencing. We thank Hugh Dingle (UC Davis) for his advice on potential migration sources around the Comoros. We thank Youki Yamasaki and Catelyn Neiman for their outstanding workmanship carrying out DNA extraction and genomic DNA library preparations. We also thank Ann-Marie Waldvogel (Senckenberg BiK-F) for very helpful discussions on cross-coalescence analyses. Collections in Zambia were supported by the NIH-funded Southern and Central Africa International Centers of Excellence for Malaria Research (2U19AI089680). This work was supported by the University of California Davis Bridge Funding, UC Davis Signature Research in Genomics Program, National Institutes of Health grant R56 AI130277, and the University of California Irvine Malaria Initiative.

## Author contributions

G.C.L., A.J.C., and Y.L. conceived the study. A.O. collected field samples in the Comoros. M.M. and D.E.N. collected field samples in Zambia. Y.L. and T.C.C. performed whole-genome sequencing. H.S. and Y.L. performed population genomic analyses. H.S., G.C.L., Y.L., M.J.H., O.D.K., T.C.C., and M.S. interpreted results. H.S. drafted the manuscript. All authors contributed to the final version of the manuscript.

## Competing interests

The authors declare no competing interests.
