## [Peer Review File · Communications Biology]

Reviewers' comments:

Reviewer #1 (Remarks to the Author):

I think this is a well written, important and interesting manuscript which is suitable for publication providing it is revised. As an aside, line numbers would have been useful.

Summary of the paper

Based on analyses of 111 *Anopheles gambiae* whole genome sequenced (WGS) samples collected across mainland Africa (Mali, Cameroon, Zambia and Tanzania) and from three of four Comoros islands (Grande Comore, Mohéli and Anjouan), this manuscript proposes a new model claiming that these important malaria vectors dispersed across Africa from West to East via a series of founder effects (Figure 6). This contrasts with previous thinking and is the title claim of the manuscript.

The important research gap that this manuscript aims to address is the current lack of analysis of *A. gambiae* WGS data from remote oceanic islands alongside *A. gambiae* WGS data from mainland Africa. This is an important gap because island populations are being considered for field trials of genetically engineered mosquitoes. Based on their analyses, the authors conclude that the Comoros islands are geographically isolated and that Grande Comore is isolated from the two smaller islands making the Comoros islands "ideal" sites for field trials of genetically engineered mosquitoes.

The analyses supporting the relative isolation of the Comoros include an admixture analysis (Figure 2b and Supplementary Figure 1); a model-based geo-genetic clustering analysis (Figure 3); a F_{ST} analysis (Figure 4); and to a lesser extent an analysis of the relative cross-coalescence (RCC) between populations (lesser since not all pairwise comparisons are shown, and not all between Comoros and mainland are ranked at the bottom). The fact that Grande Comore, Mohéli and Anjouan have lowest diversity (Figure 2a), and lowest population sizes (Figure 5a) is also consistent with their isolation. I have some reservations with the existing analyses (see suggested revisions), however the signal in the data that supports the overall conclusion appears to be strong. I believe more analyses (to establish the porosity of the islands in absolute terms) are needed to claim their suitability for field site consideration; see #1 below. Moreover, I think it's critical to recognise "ideal" has a limited sense. Many other factors that are beyond the scope of this article (e.g. consent of the local population) are critical also.

Analyses of historical population sizes and RCC support the title claim (the above mentioned clustering and diversity analyses are consistent but neither confirm nor refute a historical model of evolution). I think this is a reasonable conclusion, but its status as a model needs to be clearer to prevent oversell (see #2 below).

I agree with the overall interpretation of the F_{ST} estimates, however I am concerned about the reliability of the individual estimates. Some of estimates are based on only one (NGAB) or two (SAMA) samples per population, and at most they are based on only six (Supplementary Table 3). F_{ST} estimates based on small sample sizes are liable to be unstable, especially if population samples are collected from related individuals (as identity-by-descent analyses may reveal). In light of this and other concerns, I've suggested several revisions below.

Suggested revisions

1. All the methods used attempt to recover genetic structure due to fluctuations in allele frequencies, which accumulate relatively slowly in time. Is there any evidence of identity-by-descent between any

of the sample pairs? In other words, is there any evidence of recent migrants within the Comoros islands and between the Comoros islands and mainland Africa? This would help to estimate isolation in absolute terms - the islands may be relatively isolated but nevertheless porous (as the best fitting admixture analysis with $K = 5$ suggests).

2. Title: in light of the fact that West to East dispersal is a "proposed model (Fig. 6)" based on relatively limited analyses (Figure 5), the title claims too much. Perhaps add "a new model"? Also since the only analyses supporting the West to East Africa claim, are those of the historical population sizes and cross-coalescence", I suggest that this is clearly stated in the discussion, e.g. "From analyses of the historical population sizes and cross-coalescence, it seems..." instead of "From our data, it seems..." (Discussion, paragraph 2).

3. Mosquito DNA was sampled from adults in mainland Africa and as larvae from the Comoros (methods). As such results may be confounded by sample collection. At the very least, this is worth a cautionary remark in the discussion. Could more data be added to explore potential biases?

4. To make these results fully reproducible, code should be made available (e.g. on github).

5. Remove the F_{ST} estimates for NGAB (only one sample)

6. Consider comparing F_{ST} estimates with others computed using an estimator designed for small and unequal sample sizes (e.g. Reich, D., Thangaraj, K., Patterson, N., Price, A. L., & Singh, L. (2009). Reconstructing Indian Population History. *Nature*, 461(7263), 489–494), and comment on their stability.

7. To circumvent the small sample-size problem, consider estimating F_{ST} between main sites (i.e. Grande Comore, Mohéli, Anjouan, Mali, Cameroon, Zambia and Tanzania), not collection sites

8. Since unsupervised clustering algorithms are notoriously unstable (Pritchard, J. K., Stephens, M., & Donnelly, P. (2000). Inference of population structure using multilocus genotype data. *Genetics*, 155(2), 945–959), please comment on the stability of the Neighbor-joining tree (e.g. is the interpretation the same if estimates are based on the main sites)

9. The dispersal distances referred to in paragraph 2 of the results section do not appear to feature in any Figure or Table. Please add a plot or a table showing the inferred dispersal distances.

10. Regarding Figure 5, how robust are the trajectories? Why do some estimates (e.g. Mali vs Cameroon RCC) start and end earlier than others? And why are some comparisons (e.g. Anjouan vs Mali) not shown?

11. Please comment on how the nucleotide variability (Figure 2a) an admixture analyses (Figure 2b, especially given the best fitting scenario $K= 5$) for Grande Comore and how the position of Grande Comore on the F_{ST} -based NJ tree (Figure 3a) fits into the proposed model where Grande Comore is most recently colonised (Figure 6).

12. Why are more stringent quality control steps used in the historical analyses only (Method, bottom of page 13)?

13. The statement "However, these islands are located only ~5 km from the mainland" is inconsistent with the information in Ref 30: data from 5 islands from 4 to 50 km offshore, where one island (Banda) is 30 km offshore (Introduction, paragraph 3).

Minor revisions:

14. Keywords: I appreciate islands are important to the topic of gene drives, but this study does not directly feature “genetically engineered mosquitoes”.

15. Regards malaria cases and deaths (Introduction, paragraph 2), please ensure the “estimate” is clearly stated and the citation is complete (a more recent one is available). For example, “a disease that caused over 200 million cases and almost half a million deaths in 2017 according to WHO estimates [World malaria report 2018].

16. Since the community effect of the insecticidal nature of insecticide treated nets is sometimes under appreciated, consider revising “physical prevention like bed nets” to “physical prevention provided by bed nets” (Introduction).

17. Regards genetically engineered mosquitoes (Introduction), considering adding a review citation, e.g. Hammond et al. Pathogens and global health (2017) and the more recent Anopheles gambiae citation e.g. Kyrou et al. Nature biotechnology (2018). Similarly, could add a more focused modelling citation.

18. I disagree with the use of “highest” regarding value for assessing feasibility (Introduction, paragraph 2), and ‘ideal’ in the discussion. It is an important factor but others are too (e.g. community engagement).

19. Figure 1: reference to Supplementary Table 3 is needed to resolve the sites in Mali; add the sample counts (N, Supplementary Figure 3) after the sample site codes; please provide a reference for the CleanTOPO2 base map if available.

20. Supplementary Figure 1: it might be worth visualising as there a lot of numbers (e.g. histogram to quickly communicate mean and width and outliers)

21. Please provide a reference to justify the quantification of Genome-wide nucleotide variability (Results, paragraph 2) and change Figure 2a vertical axis label and caption to “Genome-wide nucleotide variability” for consistency with the main text.

22. To the caption of Fig. 2a, add citation for CV error analysis and reference to Supplementary Figure 2

23. Consider separating Fig 2a and 2b into Fig 2 and Fig 5, respectively.

24. Figure 3: How does the ellipse area represent the CI? Does it “represents the 95% CI width”? Add citation [37] to Figure 3 caption

25. Since distances in Figure 3 are based on a genetically updated geographic prior, “Geographic distances were not correlated to genetic distances” ought to be “Geographic distances were not correlated to geo-genetic distances” (results, paragraph 2).

26. Either parameters are estimated or estimates are computed (Results, paragraph 3), consider revision: “Fixation indices as a measure of relative population differentiation were estimated for each population pair (FST estimates, Supplementary Table 2)”.

27. Re the caption of Figure 4, I find "Long branches reflect high Fst values to distant populations" misleading since distance does not feature in the computation of the FST estimates. Please consider changing to "Long branches separate population pairs with high Fst estimates".
28. Please add confidence intervals to the Fst estimates in Figures 4b and 4c and to Supplementary table 2
29. Regarding the legend of Fig. 4b, "Within an island" is inconsistent with the main text "within each of the Comoro islands". Consider "Within islands in Comoros".
30. Regarding Fig. 4b, negative FST values are an artefact of the estimator, not biologically meaningful. The authors might consider truncating to zero as suggested and/or comment on the fact that negative values is simply an artefact of the estimator and not biologically insightful. How does this effect the stability of the NJ tree algorithm?
31. Improve resolution of all Supplementary Figures, especially Supplementary Figure 3
32. Personally, I found the paragraph on "Historical population sizes and cross-coalescence" difficult to follow because the narrative proceeds from the right to left of Figure 5. To orientate reader, a phrase could be incorporated, e.g. "Considering the trajectory of time from deep to recent past (right to left, Figure 5)" (Alternatively, the authors could switch the horizontal axis of Figure 5 such that the deep past is on the left.), and more time estimates could be added (e.g. "separating Zambia from the others ~100,000 years ago" etc.).
33. Figure 5, please align the figure the horizontal axes of a and b to facilitate comparison of effective population size and RCC
34. Figure 5 caption, please provide a citation for the geological appearance of the Comoro islands.
35. "ability" or "vagility"?
36. Where 53 is cited in the discussion, could also cite Ogaugwu, C. E., Agbo, S. O., & Adekoya, M. A. (2019). CRISPR in Sub-Saharan Africa: Applications and Education. Trends in Biotechnology, 37(3), 234–237.
37. In the Methods, Mosquito Samples, please make more clear the fact that these samples were re-sequenced (as stated in introduction, paragraph 4, page 11).
38. If geographic distances are used in the SpaceMix analysis, distances reflect "geo-genetic, rather than geographic" not "genetic, rather than geographic" (Methods, top of page 13).
39. Did the Bauwelch algorithm converge within 20 iterations? Or was this a pre-determined limit (Methods, page 14)?

Reviewer #2 (Remarks to the Author):

Reviewer

The main goal of this study was to establish the genetic relationships and gene flow between A.

gambiae populations among three Comoro Islands and four continental African countries.

The authors had an interesting insight to explore islands far away from the mainland. I consider this study of high interest for the readers. It is original, brings a new perspective of *A. gambiae* population's evolution, and shed light on the future concerns regarding the use of the gene drive mechanisms.

I judge this article as "accepted" after "minor revision". The authors can find my comments below.

Main commentaries:

- 1- Lines 37 to 83: The introduction section is well-written with all the background being synthesized and connected, and the hypothesis is raised elegantly.
- 2- Line 238: The number of samples from the mainland differs greatly between countries. Is this a relevant concern about underrepresentation that should be addressed by the authors?
- 3- Lines 241 to 245: The collecting time-scale has a range of almost 10 years (2006, 2011, 2012, and 2015). Is this a drawback for this study? How can this affect the findings? However, the Comoros were collected in Feb 2011, which is a good point.
- 4- How did the authors establish "The genetic variability was calculated for each mosquito as the number of biallelic SNP sites in heterozygous state divided by the total number of loci in the genome"? Is there a reference? Is this consensual in the field? Is this unprecedented?
- 5- The methods section has all the necessary information, and provides a guide for interested readers to perform similar study.
- 6- Line 503: The figure should be larger, especially because the the greened dots are not so highlighted from the background and the Mali samples are overlapped.
- 7- The results and discussion sections are well-written and consistent. The discussion section highlights the main findings and it really connects them to the application purposes.

Response to the reviewers

Dear reviewers,

We thank you very much for your constructive comments.

We have addressed all of the points raised, which substantially improved our manuscript.

Below you will find our responses to your suggestions printed in blue in between your statements.

We hope that the revisions will make the manuscript suitable for publication in *Communications Biology*.

Best regards,

Hanno Schmidt *et al.*

Response to Reviewers

Reviewer #1 (Remarks to the Author):

I think this is a well written, important and interesting manuscript which is suitable for publication providing it is revised. As an aside, line numbers would have been useful.

Summary of the paper

Based on analyses of 111 *Anopheles gambiae* whole genome sequenced (WGS) samples collected across mainland Africa (Mali, Cameroon, Zambia and Tanzania) and from three of four Comoros islands (Grande Comore, Mohéli and Anjouan), this manuscript proposes a new model claiming that these important malaria vectors dispersed across Africa from West to East via a series of founder effects (Figure 6). This contrasts with previous thinking and is the title claim of the manuscript.

The important research gap that this manuscript aims to address is the current lack of analysis of *A. gambiae* WGS data from remote oceanic islands alongside *A. gambiae* WGS data from mainland Africa. This is an important gap because island populations are being considered for field trials of genetically engineered mosquitoes. Based on their analyses, the authors conclude that the Comoros islands are geographically isolated and that Grande Comore is isolated from the two smaller islands making the Comoros islands “ideal” sites for field trials of genetically engineered mosquitoes.

The analyses supporting the relative isolation of the Comoros include an admixture analysis (Figure 2b and Supplementary Figure 1); a model-based geo-genetic clustering analysis (Figure 3); a F_{ST} analysis (Figure 4); and to a lesser extent an analysis of the relative cross-coalescence (RCC) between populations (lesser since not all pairwise comparisons are shown, and not all between Comoros and mainland are ranked at the bottom). The fact that Grande Comore, Mohéli and Anjoua have lowest diversity (Figure 2a), and lowest population sizes (Figure 5a) is also consistent with their isolation. I have some reservations with the existing analyses (see suggested revisions), however the signal in the data that supports the overall conclusion appears to be strong. I believe more analyses (to establish the porosity of the islands in absolute terms) are needed to claim their suitability for field site consideration; see #1 below. Moreover, I think it's critical to recognise "ideal" has a limited sense. Many other factors that are beyond the scope of this article (e.g. consent of the local population) are critical also.

Analyses of historical population sizes and RCC support the title claim (the above mentioned clustering and diversity analyses are consistent but neither confirm nor refute a historical model of evolution). I think this is a reasonable conclusion, but its status as a model needs to be clearer to prevent oversell (see #2 below).

I agree with the overall interpretation of the F_{ST} estimates, however I am concerned about the reliability of the individual estimates. Some of estimates are based on only one (NGAB) or two (SAMA) samples per population, and at most they are based on only six (Supplementary Table 3). F_{ST} estimates based on small sample sizes are liable to be unstable, especially if population samples are collected from related individuals (as identity-by-descent analyses may reveal). In light of this and other concerns, I've suggested several revisions below.

Suggested revisions

1. All the methods used attempt to recover genetic structure due to fluctuations in allele frequencies, which accumulate relatively slowly in time. Is there any evidence of identity-by-descent between any of the sample pairs? In other words, is there any evidence of recent migrants within the Comoros islands and between the Comoros islands and mainland Africa? This would help to estimate isolation in absolute terms - the islands may be relatively isolated but nevertheless porous (as the best fitting admixture analysis with $K = 5$ suggests).

We absolutely agree with the reviewer that for field trial evaluation, it would be ideal to have recent dispersal history data. However, for more accurate phasing needed for IBD segment identification, a genetic map, mapping the recombination rate (in cM) to genome coordinates, is needed but unavailable to date for this species as to our knowledge. We discussed this as a future study needed to evaluate field trial sites and it is on top of our priority list for upcoming analyses.

The Admixture $K=5$ clustering shows the uncertainty in classifying East African continental samples (Zambia, Tanzania) to either Cameroon samples or Grande Comore. Considering the other outcomes

(PCA, SpaceMix and F_{ST} analyses), we concluded that this is due to historic migration patterns that occurred around 40,000-70,000 years ago (See Fig 6) rather than recent migration. The likelihood values calculated by software are subject for interpretation (see relevant discussions in Pritchard et al. ¹).

“There are also biological reasons to be careful interpreting K. The population model that we have adopted here is obviously an idealization. We anticipate that it will be flexible enough to permit appropriate clustering for a wide range of population structures. However, as we pointed out in our discussion of data set 3 (Choice of K for simulated data), clusters may not necessarily correspond to “real” populations.”

Our previous datasets also needed to incorporate biological reasons for interpreting K and had explored Evans’s delta K or entropy into consideration ^{2,3}.

2. Title: in light of the fact that West to East dispersal is a “proposed model (Fig. 6)” based on relatively limited analyses (Figure 5), the title claims too much. Perhaps add “a new model”? Also since the only analyses supporting the West to East Africa claim, are those of the historical population sizes and cross-coalescence“, I suggest that this is clearly stated in the discussion, e.g. “From analyses of the historical population sizes and cross-coalescence, it seems...” instead of “From our data, it seems...” (Discussion, paragraph 2).

The reviewer is right - the title claim was a bit enthusiastic and has now been specified by adding the suggested phrase.

3. Mosquito DNA was sampled from adults in mainland Africa and as larvae from the Comoros (methods). As such results may be confounded by sample collection. At the very least, this is worth a cautionary remark in the discussion. Could more data be added to explore potential biases?

Thank you for pointing out the potential bias. We added a whole new paragraph in the discussion section that is dedicated to this issue. We also revised our method section to clarify potential biases in our analysis.

4. To make these results fully reproducible, code should be made available (e.g. on github).

We posted any custom scripts we used for analysis on the github page <https://github.com/travc/gambiae-dispersal>

A statement is now included; see the revised Data availability section.

5. Remove the F_{ST} estimates for NGAB (only one sample)

Done - the tree was also re-generated with the revised dataset.

6. Consider comparing FST estimates with others computed using an estimator designed for small and unequal sample sizes (e.g. Reich, D., Thangaraj, K., Patterson, N., Price, A. L., & Singh, L. (2009). *Reconstructing Indian Population History*. *Nature*, 461(7263), 489–494), and comment on their stability.

We used Hudson's F_{ST} estimator. The text has been revised to make that clear. Bhatia et al. ⁴ compared numerous estimators and concluded: "Because the Hudson estimator is not sensitive to the ratio of sample sizes and does not systematically overestimate F_{ST} , we recommend that it be used to estimate F_{ST} for pairs of populations."

Estimates of F_{ST} derived from whole genome sequence data have been shown to be accurate even with very small sample sizes (i.e. $N=2$ /population with $n \gg 1,000$ loci ⁵). The 24.1 million loci used in our analysis are well in excess of the number of loci required for an accurate assessment of F_{ST} . In addition, we evaluated various minimum read depths and missing data ratios and observed results consistent with those we report in the manuscript. Similar case of stable F_{ST} estimation on low sample size ($N < 3$) but large number of loci (> 4 million) was also presented in our previous population genomics work in *Ae. aegypti* ⁶.

7. To circumvent the small sample-size problem, consider estimating FST between main sites (i.e. Grande Comore, Mohéli, Anjouan, Mali, Cameroon, Zambia and Tanzania), not collection sites

For reasons we provided in response to #6, we do not believe this is necessary.

8. Since unsupervised clustering algorithms are notoriously unstable (Pritchard, J. K., Stephens, M., & Donnelly, P. (2000). *Inference of population structure using multilocus genotype data*. *Genetics*, 155(2), 945–959), please comment on the stability of the Neighbor-joining tree (e.g. is the interpretation the same if estimates are based on the main sites)

The tree is based on Hudson's F_{ST} estimated from whole genome data. See comment on #6 on the low sample size and stability of the F_{ST} estimates and the resulting tree generated from those estimates. We chose Neighbor-joining for its simplicity since the signal in our data does not require more powerful methods to clearly demonstrate the pattern.

9. The dispersal distances referred to in paragraph 2 of the results section do not appear to feature in any Figure or Table. Please add a plot or a table showing the inferred dispersal distances.

We have added the Raddle results to the main text.

10. Regarding Figure 5, how robust are the trajectories? Why do some estimates (e.g. Mali vs Cameroon RCC) start and end earlier than others? And why are some comparisons (e.g. Anjouan vs Mali) not shown?

We understand the confusion the reviewer had with the interpretation of some aspects of the coalescence results in Figure 5 (now Fig. 6). First, we'd like to make clear that we stuck to the layout (e.g. axis orientation) consistent with how all other previously published studies using this method presented this kind of data. This also includes, that we quality-filtered the MSMC2 results as suggested in literature and described in our Methods section: "All time slices with $\lambda < 5$ were excluded to account for possible phasing error affecting resolution⁷²" (last sentence of Methods section). For example, Mali N_e estimates do not "end early" but rather we do not have the estimate of recent (< 10,000 years) years. This is because MSMC2 measures the time to the first coalescence between all pairs of haplotypes. Due to increased nucleotide diversity in Mali and Cameroon populations, it has to go further back to measure the first coalescence time than between Comoros samples.

The original method PSMC developed by Li and Durbin⁷ was cited >1000 times to date and the upgraded version MSMC by Schiffels and Durbin⁸ used here is cited > 400 times to date. This is a well-established method for population size estimation and demographic history based on coalescence signals for wide range of organisms. The method has been demonstrated to work for small sample size (N=1-4) without requiring a model with specific bottlenecks, hard population splits and fixed population sizes by older methods based on allele frequencies (See discussion in⁸).

Showing all possible population combinations would have made the figure even harder to understand. Therefore, we restricted it to a subset that covers the highly relevant comparisons, omitting some far-distance couples like Anjouan vs Mali. We hope the reviewers agree with the decision.

11. Please comment on how the nucleotide variability (Figure 2a) an admixture analyses (Figure 2b, especially given the best fitting scenario K= 5) for Grande Comore and how the position of Grande Comore on the F_{ST} -based NJ tree (Figure 3a) fits into the proposed model where Grande Comore is most recently colonised (Figure 6).

The reviewer is right with pointing to the fact that the results show slightly differing topologies in different analyses. This can be attributed to varying weightings of different aspects of the data by various programs. However, we believe the different analyses' results put together a pretty clear picture. The nucleotide variability plot is undecided at most in our view – Grande Comore shows lower variability than all mainland populations and is equal to Mohéli. The fact that Anjouan has the lowest value here and thereby differs from Mohéli is a singular case and probably not meaningful given the other analyses showing very close genetic relation between the two smaller islands' populations. The mentioned F_{ST} tree and Admixture plot on the other hand explicitly support (1) the idea that Grande Comore has been colonized more recently because its population is closer related to mainland sites than are populations from the two smaller islands and (2) the idea that Grande Comore was colonized independently from mainland because its population is genetically closer to mainland than to Mohéli/Anjouan populations. We have clarified this in the main text.

12. Why are more stringent quality control steps used in the historical analyses only (Method, bottom of page 13)?

Missing data affects the phasing of genotypes into haplotypes (by imputing); the filtering is a typical procedure for this kind of analysis.

13. The statement “However, these islands are located only ~5 km from the mainland” is inconsistent with the information in Ref 30: data from 5 islands from 4 to 50 km offshore, where one island (Banda) is 30 km offshore (Introduction, paragraph 3).

Number revised to <50.

Minor revisions:

14. Keywords: I appreciate islands are important to the topic of gene drives, but this study does not directly feature “genetically engineered mosquitoes”.

We removed the key word.

15. Regards malaria cases and deaths (Introduction, paragraph 2), please ensure the “estimate” is clearly stated and the citation is complete (a more recent one is available). For example, “a disease that caused over 200 million cases and almost half a million deaths in 2017 according to WHO estimates [World malaria report 2018].

Done

16. Since the community effect of the insecticidal nature of insecticide treated nets is sometimes under appreciated, consider revising “physical prevention like bed nets” to “physical prevention provided by bed nets” (Introduction).

Done

17. Regards genetically engineered mosquitoes (Introduction), considering adding a review citation, e.g. Hammond et al. Pathogens and global health (2017) and the more recent Anopheles gambiae citation e.g. Kyrou et al. Nature biotechnology (2018). Similarly, could add a more focused modelling citation.

We followed the recommendation and added the citations to the revised introduction.

18. I disagree with the use of “highest” regarding value for assessing feasibility (Introduction, paragraph

2), and 'ideal' in the discussion. It is an important factor but others are too (e.g. community engagement).

We agree and changed "highest" to "important".

19. Figure 1: reference to Supplementary Table 3 is needed to resolve the sites in Mali; add the sample counts (N, Supplementary Figure 3) after the sample site codes; please provide a reference for the CleanTOPO2 base map if available.

We added a second detail magnification for the Mali sites to the figure. The sample counts are now given in the figure caption.

20. Supplementary Figure 1: it might be worth visualising as there a lot of numbers (e.g. histogram to quickly communicate mean and width and outliers)

We assume the reviewer refers to Supplementary Table 1. We have added mean, median, max and min values to the table.

21. Please provide a reference to justify the quantification of Genome-wide nucleotide variability (Results, paragraph 2) and change Figure 2a vertical axis label and caption to "Genome-wide nucleotide variability" for consistency with the main text.

The calculation gives an estimate for the SNP density and is similar to "one SNP every X bp". However, including all SNPs would have arbitrarily weighted the similarity/differences to the reference genome (i.e. isolated populations that are completely homozygous due to founder effects or bottle necks would still get a high genetic variability estimate when several fixed positions differ from the reference genome; since this reference genome was not chosen for biological reasons but randomly, we wanted to exclude this effect) – this rationale is now given in the Methods section.

We used "genetic variability" in all but one case. We decided to stick to the wording and rather adapted the single occurrence of "genome-wide nucleotide variability".

22. To the caption of Fig. 2a, add citation for CV error analysis and reference to Supplementary Figure 2

The CV error analysis is part of the ADMIXTURE package and called "cross-validation procedure". We specify this now in the Methods section and reference the article in the figure caption.

23. Consider separating Fig 2a and 2b into Fig 2 and Fig 5, respectively.

We initially tried to keep number of items as low as possible. Since *Communications Biology* allows up to ten items, we now follow the reviewer's recommendation.

24. Figure 3: How does the ellipse area represent the CI? Does it "represents the 95% CI width"? Add citation [37] to Figure 3 caption

The ellipses represent "the 95% CI for location where an individual could have originated in geogenetic space" – we have now specified this in the figure caption. We also added the requested citation.

25. Since distances in Figure 3 are based on a genetically updated geographic prior, "Geographic distances were not correlated to genetic distances" ought to be "Geographic distances were not correlated to geo-genetic distances" (results, paragraph 2).

Done

26. Either parameters are estimated or estimates are computed (Results, paragraph 3), consider revision: "Fixation indices as a measure of relative population differentiation were estimated for each population pair (FST estimates, Supplementary Table 2)".

Done

27. Re the caption of Figure 4, I find "Long branches reflect high Fst values to distant populations" misleading since distance does not feature in the computation of the FST estimates. Please consider changing to "Long branches separate population pairs with high Fst estimates".

Done

28. Please add confidence intervals to the Fst estimates in Figures 4b and 4c and to Supplementary table 2

Done

29. Regarding the legend of Fig. 4b, "Within an island" is inconsistent with the main text "within each of the Comoro islands". Consider "Within islands in Comoros".

Done

30. Regarding Fig. 4b, negative FST values are an artefact of the estimator, not biologically meaningful. The authors might consider truncating to zero as suggested and/or comment on the fact that negative values is simply an artefact of the estimator and not biologically insightful. How does this effect the stability of the NJ tree algorithm?

Thanks for the suggestion. We changed the negative values to zero and recalculated the tree. We also use 100 different random subsets of 50K SNPs for distance matrix, calculated the consensus in each branching point and added the values to the revised tree.

Of note, all branches had over 57%. Due to tight spacing between nodes, only the bootstrap values ≥ 98 were displayed in the figure.

31. Improve resolution of all Supplementary Figures, especially Supplementary Figure 3

The original document resolution is good and legible but the compiled pdf for review lowered the resolution.

32. Personally, I found the paragraph on “Historical population sizes and cross-coalescence” difficult to follow because the narrative proceeds from the right to left of Figure 5. To orientate reader, a phrase could be incorporated, e.g. “Considering the trajectory of time from deep to recent past (right to left, Figure 5)” (Alternatively, the authors could switch the horizontal axis of Figure 5 such that the deep past is on the left.), and more time estimates could be added (e.g. “separating Zambia from the others ~100,000 years ago” etc.).

Although all cross-coalescence analysis plots are organized in the same way (deep to recent past from right to left) we agree with the reviewer that not every reader might be familiar with it. We accordingly added the suggested explanation to the results section as well as to the MSMC2 plot’s caption.

33. Figure 5, please align the figure the horizontal axes of a and b to facilitate comparison of effective population size and RCC

Done

34. Figure 5 caption, please provide a citation for the geological appearance of the Comoro islands.

Done

35. “ability” or “vagility”?

We really mean “vagility”. The mosquitoes don’t move very far independently.

36. Where 53 is cited in the discussion, could also cite Ogaugwu, C. E., Agbo, S. O., & Adekoya, M. A. (2019). CRISPR in Sub-Saharan Africa: Applications and Education. Trends in Biotechnology, 37(3), 234–237.

Done

37. In the Methods, Mosquito Samples, please make more clear the fact that these samples were re-sequenced (as stated in introduction, paragraph 4, page 11).

With resequencing we don't mean that we already sequenced the specimens before and now sequenced them again. Resequencing is also the official term when sequencing the genome *of a species that already has been sequenced* and comparing the resulting reads to known sequence data (by mapping). Since there is a reference genome for *An. gambiae* that we mapped our reads against, this is by definition resequencing (in opposite to *de novo* sequencing).

This is the first time these specimens were whole genome sequenced and the presented manuscript is the first one to report results on them.

38. If geographic distances are used in the SpaceMix analysis, distances reflect "geo-genetic, rather than geographic" not "genetic, rather than geographic" (Methods, top of page 13).

Done

39. Did the Bauwelch algorithm converge within 20 iterations? Or was this a pre-determined limit (Methods, page 14)?

This was indeed a pre-determined limit as suspected by the reviewer.

Reviewer #2 (Remarks to the Author):

Reviewer

The main goal of this study was to establish the genetic relationships and gene flow between *A. gambiae* populations among three Comoro Islands and four continental African countries.

The authors had an interesting insight to explore islands far away from the mainland. I consider this study of high interest for the readers. It is original, brings a new perspective of *A. gambiae* population's evolution, and shed light on the future concerns regarding the use of the gene drive mechanisms.

I judge this article as "accepted" after "minor revision". The authors can find my comments below.

Main commentaries:

1- Lines 37 to 83: The introduction section is well-written with all the background being synthesized and connected, and the hypothesis is raised elegantly.

Thank you.

2- Line 238: The number of samples from the mainland differs greatly between countries. Is this a relevant concern about underrepresentation that should be addressed by the authors?

Please see comment for Reviewer 1 at #6 regarding the sample size related question.

3- Lines 241 to 245: The collecting time-scale has a range of almost 10 years (2006, 2011, 2012, and 2015). Is this a drawback for this study? How can this affect the findings? However, the Comoros were collected in Feb 2011, which is a good point.

We agree that the collection span could be a sensitive issue when trying to estimate dispersal range within a relative short time (within a year). However, the degree of separation (10s of thousands of years) we identified between populations from different geographic regions, 10 years of separation do not affect the demographic history.

4- How did the authors establish “The genetic variability was calculated for each mosquito as the number of biallelic SNP sites in heterozygous state divided by the total number of loci in the genome”? Is there a reference? Is this consensual in the field? Is this unprecedented?

Please see comment for Reviewer 1 at #21.

5- The methods section has all the necessary information, and provides a guide for interested readers to perform similar study.

We are glad that our method section is to your satisfaction.

6- Line 503: The figure should be larger, especially because the the greened dots are not so highlighted from the background and the Mali samples are overlapped.

We included additional insert to zoom in for the Mali locations in Figure 1. We used more visible outline for the markers for the Dar es Salaam location to be more visible. We keep the color scheme to be the same as it is related to the other result figures throughout the manuscript.

7- The results and discussion sections are well-written and consistent. The discussion section highlights the main findings and it really connects them to the application purposes.

Thank you.

Literature

- 1 Pritchard, J. K., Stephens, M. & Donnelly, P. Inference of population structure using multilocus genotype data. *Genetics* **155**, 945-959 (2000).
- 2 Lanzaro, G. C., Collier, T. C. & Lee, Y. Defining Genetic, Taxonomic, and Geographic Boundaries Among Species of the *Psorophora confinnis* (Diptera: Culicidae) Complex in North and South America. *J Med Entomol* **52**, 907-917, doi:10.1093/jme/tjv084 (2015).
- 3 Marsden, C. D. *et al.* Asymmetric introgression between the M and S forms of the malaria vector, *Anopheles gambiae*, maintains divergence despite extensive hybridization. *Mol Ecol* **20**, 4983-4994, doi:10.1111/j.1365-294X.2011.05339.x (2011).
- 4 Bhatia, G., Patterson, N., Sankararaman, S. & Price, A. L. Estimating and interpreting FST: the impact of rare variants. *Genome Res.* **23**, 1514-1521 (2013).
- 5 Nazareno, A. G., Bemmels, J. B., Dick, C. W. & Lohmann, L. G. Minimum sample sizes for population genomics: an empirical study from an Amazonian plant species. *Mol Ecol Resour* **17**, 1136-1147, doi:10.1111/1755-0998.12654 (2017).
- 6 Lee, Y. *et al.* Genome-wide divergence among invasive populations of *Aedes aegypti* in California. *BMC Genomics* **20**, 204, doi:10.1186/s12864-019-5586-4 (2019).
- 7 Li, H. & Durbin, R. Inference of human population history from individual whole-genome sequences. *Nature* **475**, 493-496 (2011).
- 8 Schiffels, S. & Durbin, R. Inferring human population size and separation history from multiple genome sequences. *Nat. Genet.* **46**, 919-925 (2014).

REVIEWERS' COMMENTS:

Reviewer #1 (Remarks to the Author):

The authors have largely addressed my comments. Thank you for clarifying the estimator of F_{ST} used. I agree that it is appropriate for small and unequal sample sizes. Also, thank you for explaining "resequencing". I have a few outstanding comments.

1. I appreciate the reason for counting het SNPs vs all SNPs. Nevertheless, please add a reference as requested by both reviewers to justify the calculation of genetic variability in this way versus, say, nucleotide diversity (Nei and Li. PNAS 76.10 (1979): 5269-5273).

2. I strongly suggest changing "that are genetically and physically isolated" to "that are relatively genetically and physically isolated" (line 244).

3. To avoid confusion, I suggest changing "by genotyping Divergence Island SNPs..." (line 235) to "by genotyping Divergence Island SNPs prior to whole genome sequencing...".

4. I appreciate the decision to postpone analyses of identity-by-descent (IBD). If/when you come to do so, it may be helpful to know that it is possible to estimate relatedness (i.e. probability of IBD) between diploids without a genetic map and using only modest data types; see Weir et al. "Genetic relatedness analysis: modern data and new challenges." Nature Reviews Genetics 7.10 (2006): 771.

5. In my initial review I wrote "Please add confidence intervals to the F_{st} estimates in Figures 4b and 4c and to Supplementary table" and authors responded "Done" but there are no confidence intervals in revised Fig 4.

6. Regards point #11 in my initial review, I agree that 1) the F_{ST} tree ordering Mali > Cameroon > Zambia > Tanzania is consistent with dispersal from West to East via serial founder events over time; that 2) the F_{ST} and Admixture results suggest Grande Comore is more closely related to mainland than Mohéli and Anjouan; and that 3) the population size result suggests Grande Comore was the more recently colonized. However, under the West to East model, I disagree that 2) implies recent colonization. Is it not more correct to say that island colonization might depart from the West to East model? Please see the diagram in the attached file.

Below we answer point by point to the comments of the reviewer. Our answers are highlighted in blue.

REVIEWERS' COMMENTS:

Reviewer #1 (Remarks to the Author):

The authors have largely addressed my comments. Thank you for clarifying the estimator of F_{ST} used. I agree that it is appropriate for small and unequal sample sizes. Also, thank you for explaining "resequencing". I have a few outstanding comments.

1. I appreciate the reason for counting het SNPs vs all SNPs. Nevertheless, please add a reference as requested by both reviewers to justify the calculation of genetic variability in this way versus, say, nucleotide diversity (Nei and Li. PNAS 76.10 (1979): 5269-5273).

We have now included a citation (Samuels *et al.* 2016, Genetics) who present an almost identical measure.

2. I strongly suggest changing "that are genetically and physically isolated" to "that are relatively genetically and physically isolated" (line 244).

Done

3. To avoid confusion, I suggest changing "by genotyping Divergence Island SNPs..." (line 235) to "by genotyping Divergence Island SNPs prior to whole genome sequencing..."

Done

4. I appreciate the decision to postpone analyses of identity-by-descent (IBD). If/when you come to do so, it may be helpful to know that it is possible to estimate relatedness (i.e. probability of IBD) between diploids without a genetic map and using only modest data types; see Weir et al. "Genetic relatedness analysis: modern data and new challenges." Nature Reviews Genetics 7.10 (2006): 771.

Thank you for pointing at this interesting approach.

5. In my initial review I wrote “Please add confidence intervals to the F_{ST} estimates in Figures 4b and 4c and to Supplementary table” and authors responded “Done” but there are no confidence intervals in revised Fig 4.

We have added standard deviation values to all F_{ST} estimates in the Supplementary Table 2. Confidence intervals are too small to be clearly visible in Figure 4b and 4c, so we omit them.

6. Regards point #11 in my initial review, I agree that 1) the FST tree ordering Mali > Cameroon > Zambia > Tanzania is consistent with dispersal from West to East via serial founder events over time; that 2) the FST and Admixture results suggest Grande Comore is more closely related to mainland than Mohéli and Anjouan; and that 3) the population size result suggests Grande Comore was the more recently colonized. However, under the West to East model, I disagree that 2) implies recent colonization. Is it not more correct to say that island colonization might depart from the West to East model? Please see the diagram in the attached file.

We appreciate this thought. The reviewer is right; we cannot tell for sure whether Grande Comore was colonized from the two smaller islands or from mainland. We acknowledge this now in the discussion section.